# The Use of Deep Learning to Predict Stroke Patient Mortality

**DOI:** 10.3390/ijerph16111876

**Published:** 2019-05-28

**Authors:** Songhee Cheon, Jungyoon Kim, Jihye Lim

**Affiliations:** 1Department of Physical Therapy, Youngsan University, Yangsan 626-790, Korea; 1000sh@ysu.ac.kr; 2Department of Computer Science, Kent State University, Kent, OH 44242, USA; jykim2@kent.edu; 3Department of Healthcare Management, Youngsan University, Yangsan 626-790, Korea

**Keywords:** stroke, prediction, deep learning, feature extraction

## Abstract

The increase in stroke incidence with the aging of the Korean population will rapidly impose an economic burden on society. Timely treatment can improve stroke prognosis. Awareness of stroke warning signs and appropriate actions in the event of a stroke improve outcomes. Medical service use and health behavior data are easier to collect than medical imaging data. Here, we used a deep neural network to detect stroke using medical service use and health behavior data; we identified 15,099 patients with stroke. Principal component analysis (PCA) featuring quantile scaling was used to extract relevant background features from medical records; we used these to predict stroke. We compared our method (a scaled PCA/deep neural network [DNN] approach) to five other machine-learning methods. The area under the curve (AUC) value of our method was 83.48%; hence; it can be used by both patients and doctors to prescreen for possible stroke.

## 1. Introduction

Stroke is one of the leading causes of death and disability worldwide [1]. In Korea, stroke is the second-leading cause of death [2]. The Korean population is aging very rapidly; the percentage aged ≥ 60 years is predicted to increase from 13.7% in 2015 to 28.6% by 2050 [3]. Incidence of stroke increases with age. Stroke patients have longer hospital stays and higher re-admission rates and medical costs than patients with other chronic diseases [4,5]. In the U.S., the total annual direct medical costs of stroke in 2008 were USD 18.8 billion; in the same year, the per-person expenditure was USD 7657 [6]. Length of hospital stay, medical expenditure, readmission rate, and accompanying comorbidities greatly affect medical resource use [7,8,9,10]. Studies on stroke patients are very important for efficient utilization of medical resources.

The concept of artificial intelligence (AI) has recently permeated various sectors of life, including rapidly evolving healthcare systems [11]. As electronic diagnoses, therapies, and record-keeping expand, it is essential to leverage, integrate, and optimize these advances [12]. In the field of medicine, patient data are amassed in distributed electronic health record (EHR) databases and voluminous clinical, imaging, and laboratory datasets, among others [13]. Such data can be utilized to predict disease incidence and prognosis.

Recent nationwide efforts seek to use big data to expand precision medicine to many other medical areas [14,15]. Precision medicine is broadly defined as patient-specific diagnosis and therapy [16]. EHRs and health insurance claims data can aid precision medicine by improving prognostic models [11]. Deep learning using big data has been employed to predict disease [17,18,19]. Deep learning is actively used in many fields, yielding satisfactory results when conventional analyses are not appropriate [20,21]. The deep-learning model of Xu et al. afforded better predictive performance than a generalized linear model (GLM), a least absolute shrinkage and selection operator (LASSO) model, and an autoregressive integrated moving average (ARIMA) model [22]. Therefore, deep learning can predict disease. However, few studies have sought to predict stroke mortality using big data.

To date, there have not been deep learning-based, but heuristic or nature-inspired methods for detecting stroke or cardiovascular diseases. Teoh [23] applied the neural network using different sources of temporal data from the electronic health record through a dual-input topology. Although this study used statistical data to predict stroke, major risk factors related to stroke were missed. Pereira et al. [24] provide stroke detection system using convolutional neural network with computed tomography. Although overall detection accuracy of this study is relatively high, it needs detailed medical images to diagnose the occurrence of stroke. Beriteli et al. [25] proposed a training technique to diagnose the ECG signals using the neural network. Wu et al. [26] applied the neural network to assess the risk levels of hypertension with health examination data. 

Here, we identify factors affecting stroke mortality, and derive a predictive model based on deep learning, employing 2013–2016 Korean National Hospital Discharge In-depth Injury Survey (KNHDS) data. This will allow healthcare policymakers to improve the quality of medical care, evaluate its appropriateness, and employ diagnostic resources efficiently. Our research was performed to predict stroke mortality using large-scale electronic health records. This study is expected to expand the research that can prescreen diverse diseases the in e-health field in future.

## 2. Related Work

Several studies have used deep learning methods to solve various problems [27,28,29,30,31]. In particular, there have been many computer-aided diagnosis systems using deep learning for detecting diverse diseases [32,33,34,35,36,37,38,39,40,41]. Machine-learning/deep learning has been employed to detect or predict certain diseases using various approaches and datasets. Kim et al. [42] developed and validated several machine-learning models (i.e., support vector machine—SVM, random forest—RF, artificial neural network—ANN, and linear regression—LR) to identify the risk of osteoporosis in postmenopausal women; the cited authors used medical records such as those of the Korean National Health and Nutrition Surveys. Although its accuracy was acceptable, only relatively small datasets were used (1000 patients for training and 674 for testing). Wang et al. [43] sought to detect heart failure earlier using structured and unstructured data from EHRs and an RF classifier. Arandjelović et al. [44] applied a Markov process to predict various health outcomes using electronic medical records (EMRs). Putin et al. [45] sought to predict human chronological age using deep neural networks (DNNs); 60,600 common blood biochemistry and cell count test results were evaluated. Yoo et al. [46] developed a self-assessment system identifying adults at high risk of knee osteoarthritis using an ANN and various datasets. Hung et al. [19] used several classifiers (i.e., DNN, gradient-boosting decision tree—GBDT, LR, and SVM) to explore an EMC database of about 800,000 patients; the aim was to predict five-year stroke occurrence. Rajkomar et al. [47] used deep learning to predict several medical events by analyzing EMR data (216,221 adult patients). All of these previous studies used machine-learning/deep learning to detect diverse diseases. However, to the best of our knowledge, there have been few efforts to predict stroke with these methods. 

## 3. Materials and Methods

### 3.1. Subjects

We used data from the KNHDS, collected from 2013 to 2016 by the Korea Centers for Disease Control and Prevention (KCDC) (Figure 1). The KNHDS collected data from about 150 hospitals nationwide, all with more than 100 beds [48]. The subjects were 15,099 stroke patients with primary International Classification of Diseases diagnostic codes corresponding to hemorrhagic stroke (I60–I62) and ischemic stroke (I63). Table 1 shows general statistical information about patients. The mean age of the subjects was 66 years. Of the patients, 54.7% were male, 45.3% were female; and 6.9% (1038 people) were patients who died.

### 3.2. Principal Variables

The dependent variable was mortality rate of stroke patients. Independent variables reflecting social demographic status included gender, age, and type of insurance. Medical variables included mode of admission, length of hospital stay, hospital region, total number of hospital beds, stroke type, brain surgery status, and Charlson Comorbidity Index (CCI) score. The CCI is widely used to adjust for comorbidities, and is given by the sum of weighted scores based on the presence/absence of 19 different medical conditions [49]. However, we excluded cerebrovascular diseases because they may overlap with the primary disease. Brain surgery was defined as microvascular decompression, craniotomy, cranioplasty, ventriculostomy with shunting, removal of a subdural/epidural hematoma, and endarterectomy. Type of insurance was defined as national health insurance, medicare, industrial accident, and car insurance. Admission mode was defined as emergency, ambulatory, and others. Stroke type was categorized as ischemic stroke and hemorrhagic stroke. The hospital regions were Seoul, metropolitan Seoul, Gyeonggi, and other. The total number of hospital beds per region was 100–299, 300–499, 500–999, or ≥ 1000.

### 3.3. Methods

Our deep learning model included 11 variables: gender, age, type of insurance, mode of admission, brain surgery required, region, length of hospital stay, hospital location, the number of hospital beds, stroke type, and the CCI. We used a DNN/scaled principal component analysis (PCA) to automatically generate features from the data and identify risk factors for stroke. We enrolled 15,099 subjects with a history of stroke. Figure 2 shows the system architecture: (1) We used Korea National Health and Nutrition Examination Survey (KNHANES) data for the 11 included variables, where these data were divided into training (66%) and testing sets (34%); in the training set, we used 30% of the samples for validation; (2) we preprocessed data using both PCA and a scaler to convert categorical variables into continuous variables, and to generate models for testing; (3) we then trained the DNN using the scaled PCA variables and (4) compared the predicted results to “ground truth” data (clinician labels). The training and test data did not overlap. 

Traditionally, detection performance is measured by evaluating accuracy (*Acc*). However, an imbalance was evident in the stroke datasets; more survival (*n* = 13,971) than non-survival (*n* = 1038) data were present. Thus, we used three additional metrics: sensitivity (*Sn*), specificity (*Sp*), and positive predictive value (*PPV*). The *Sn* reflects the probability of detecting non-survival; *Sp* reflects the probability of detecting survival; and PPV is the probability that non-survival status was corrected for appropriately. We used four parameters, true-positive (*TP*), true-negative (*TN*), false-positive (*FP*) and false-negative (*FN*), to evaluate model performance. The *TP* is the correctly predicted stroke rate and the *TN* is the correctly predicted non-stroke rate. The *FP* and *FN* are the incorrectly predicted stroke and non-stroke rates, respectively. The *Sn*, *Sp*, *PPV*, and *Acc* were calculated as follows:*Sn* = *TP*/(*TP* + *FN*)(1)
Sp = *TN*/(*TN* + *FP*)(2)
*PPV* = *TP*/(*TP* + *FP*)(3)
*Acc* = (*TP* + *TN*)/(*TP* + *FN* + *FP* + *TN*)(4)

### 3.4. Preprocessing

PCA is a simple non-parametric method used to extract useful information from elaborate datasets. In general, PCA preprocessing efficiently defines new features, reducing dimensions to find hidden or simplified structures for inclusion in classification algorithms [50]. However, the dataset that we used featured major categorical/binary and minor continuous variables. The former variables should not be input into a DNN classifier because they lack detailed information. We used the PCA maximum-attribute filter to convert all 11 binary or categorical variables into 11 continuous variables to minimize data discretization. Figure 3 shows the first and second principal components of four different PCAs: (a) a normal PCA; (b) a PCA with a standard scaler; (c) a PCA with a min/max scaler; and (d), a PCA with a quantile transformer scaler. A DNN/PCA-quantile-transformer scaler afforded the best performance.

### 3.5. DNN Architecture

We employed simple feed-forward neural networks, trained using a standard backpropagation algorithm, in our deep (four hidden layers) learning models. For each DNN, we adjusted several hyperparameters, including the number of hidden layers, the number of neurons in each layer, the activation function, the optimization method, and the regularization technique. The best DNN featured four hidden layers with 22, 10, 10, and 10 neurons, respectively. The last layer, with one neuron, yielded a regression output. Accuracy served as the optimization loss function (we applied regularization terms). The DNN featured ReLU activation [51] in each layer; the dropout [52] probability was 0.2 for each layer. We used Adam optimization [53] with 0.001 for learning rate and L2 regularization during training; this optimizer is robust in terms of hyperparameter choice and, empirically, has shown very good performance. We applied batch normalization [54] after the first two layers to counter overfitting and ensure stable convergence. Figure 4 shows the architecture of the proposed DNN.

All models were implemented using Keras [55] with a TensorFlow [56] backend. Binary cross entropy served as the loss function when evaluating stroke development in Figure 5. As the dataset classes were not balanced, we applied class weighting; this rendered minority classes more significant. 

## 4. Results and Discussion

Of the various settings tested, the best DNN architecture featured four hidden layers, each with 22 or 10 neurons, 50 training epochs, and a batch size of 5. Table 2 shows the confusion matrix resulting from scaled PCA preprocessing. 

Table 3 summarizes the computational results yielded by the classification algorithms; we list thresholds, confusion matrix values, and five performance parameters; the AUCs of the top two classifiers are highlighted in bold. Considering all model parameters, the optimal stroke probability threshold was 0.13, with a model *Acc* of 84.03%, *Sn* of 64.32%, *Sp* of 85.56%, and *PPV* of 25.7%. We determined the thresholds of each classifier to make the balance between the sensitivity and specificity empirically. Compared to the commonly used performance metrics (*Sn, Sp, PPV,* and *Acc*), the area under the curve (AUC; a single value) better reflects algorithm performance [57]. Our method afforded an AUC of 83.48%; a comparison of the receiver operating characteristic (ROC) curve (indicating the predictive performance of our DNN/scaled PCA model) and the ROC curves of other classifiers is shown in Figure 6. The DNN/scaled PCA algorithm was optimal, followed by the AdaBoost classifier. Table 4 shows the comparison of the performance using 10-fold cross validation in order to verify the results from Figure 6 and Table 3. The AUCs of the top two classifiers are highlighted in bold, such as DNN and ADB. These results clearly support the conclusion that the DNN/scaled PCA algorithm outperforms the other five algorithms.

We used the DNN/scaled PCA classifier to estimate stroke occurrence, and derived correlation coefficients between various patient variables and stroke. However, these did not clearly reveal the relationships between principal components and stroke. Table 5 lists the four correlation coefficients (r-values; over +/− 0.09) among the 11 variables; the best correlation (+0.2) is highlighted in bold and other informative results (over +/− 0.09) are in italics and shown in red.

We considered tuning hyperparameters, such as the number of nodes and depth of the DNN, to improve stroke detection. Although tuning can be valuable, no general rule is available; we would have had to train 2–8 layers with 10–40 nodes based on trial-and-error. We used two methods to prevent overfitting, dropout and batch normalization; the latter prevents loss of feed-forward data in a manner similar to appropriate weighting on initialization, and dropout uses weighting to minimize the effects of certain hidden nodes. 

Our method predicts stroke using indirect or limited data, such as medical service use history and health behavior. This will reduce future medical costs and facilitate diagnosis. The limitations of our work included a lack of input data separation and the lack of longitudinal data. Also, we employed survey data, which has drawbacks which include the binary format. Although we used scaled PCA to improve the data resolution, additional input variables may be required. In addition, our results apply only to subjects who may suffer from stroke in the future; we excluded those currently receiving treatment for stroke. This may reduce the overall accuracy of our predictive model.

## 5. Conclusions

Based on the data of 15,099 subjects, we developed a deep learning model featuring scaled PCA to automatically predict stroke based on medical utilization history and health behaviors. No subjective variables were included in the model. Our work allows early detection of patients at high risk of stroke who need additional checkups and appropriate treatment prior to disease exacerbation. Our method renders it unnecessary to select variables manually. As the input data are simple (albeit of low resolution, that is, binary or with a limited number of choices), we used a DNN to study the variables of interest and scaled PCA to generate improved continuous inputs for the DNN. The sensitivity, specificity and AUC value of our method were 64.32%, 85.56% and 83.48%, respectively. Our method can be used not only to predict stroke using limited data, but also other diseases.

In future, we will modify and apply our method for the analysis of other medical service use and health behavior datasets on conditions such as dementia. We will also use detailed indices and physiological signals as input data to achieve more meaningful DNN results. Finally, we will employ auto-fine-tuning methods to reduce training time and improve performance.

## Figures and Tables

**Figure 1 ijerph-16-01876-f001:**
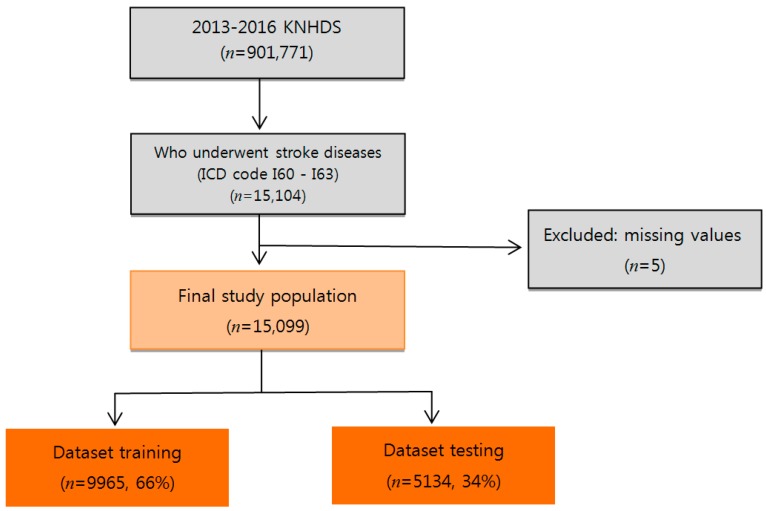
The patient selection process. KNHDS = Korean National Hospital Discharge In-depth Injury Survey. ICD = International Classification of Diseases.

**Figure 2 ijerph-16-01876-f002:**
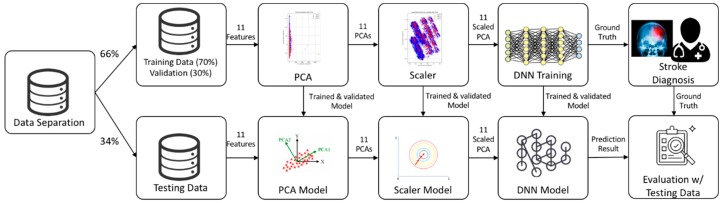
The architecture of the deep neural network (DNN)/scaled principal component analysis (PCA) approach.

**Figure 3 ijerph-16-01876-f003:**
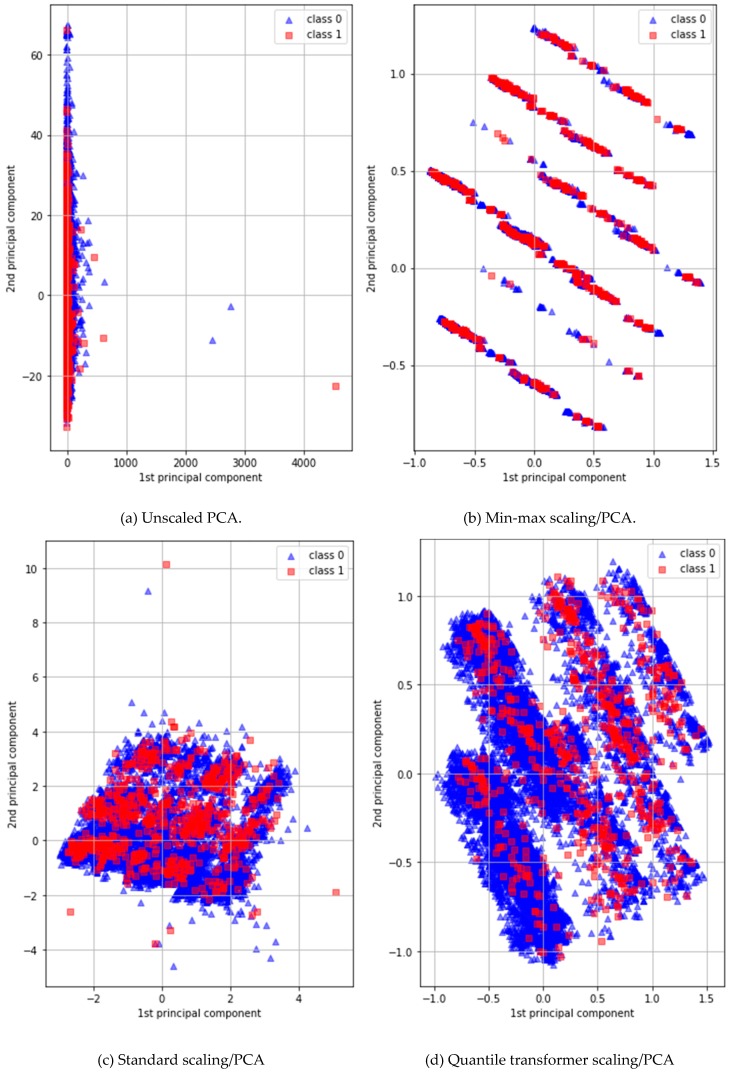
Two-dimensional plots of the first and second principal components (class 0 indicates non-stroke patients and class 1 indicates stroke patients).

**Figure 4 ijerph-16-01876-f004:**
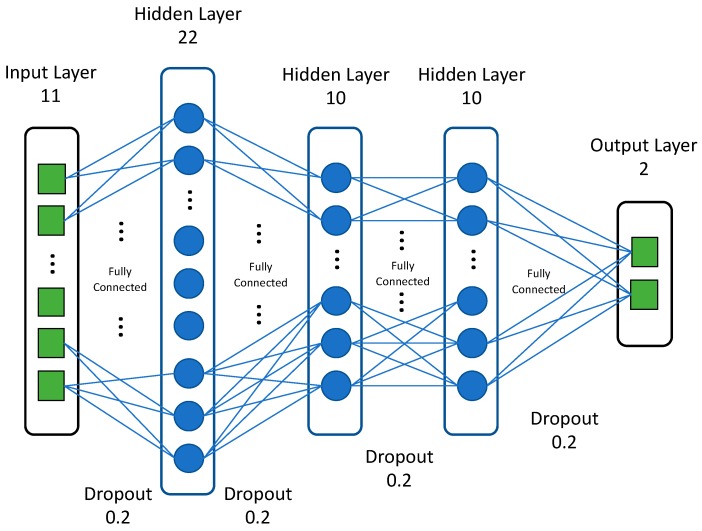
The architecture of the proposed DNN.

**Figure 5 ijerph-16-01876-f005:**
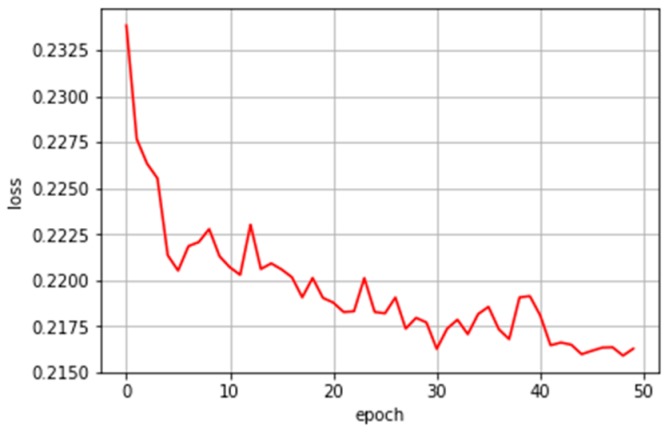
Train loss during training with early stopping.

**Figure 6 ijerph-16-01876-f006:**
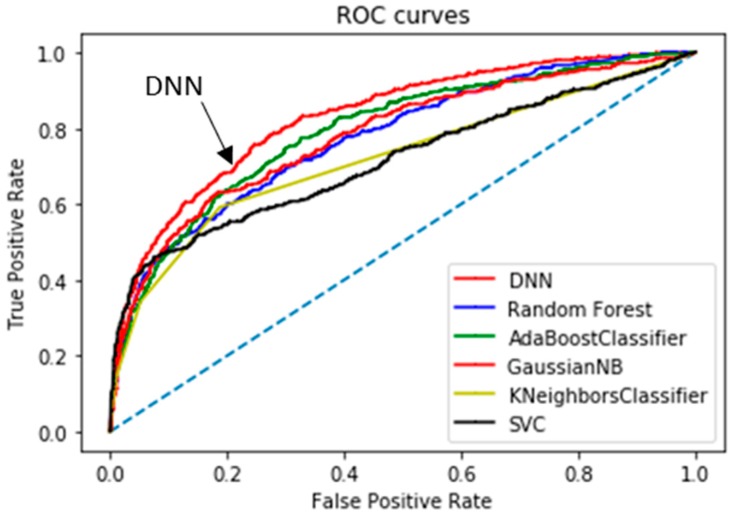
Regarding predictive performance, the area under the receiver operating characteristic curve value was highest (83.48%) for the DNN/scaled PCA classifier.

**Table 1 ijerph-16-01876-t001:** Distribution of subjects by general characteristics.

Variables		N (%)
Mean age		66.1 years
Gender	Male	8252 (54.7)
	Female	6847 (45.3)
Mortality	Yes	1038 (6.9)
	No	14,061 (93.1)
Stroke type	ischemic	10,668 (70.7)
	hemorrhagic	4431 (29.3)

**Table 2 ijerph-16-01876-t002:** Confusion matrix for our method.

Confusion Matrix	Predicted (T)	Predicted (F)
Actual (T)	238	132
Actual (F)	688	4076

**Table 3 ijerph-16-01876-t003:** Comparison of the confusion matrix values and performance for six classifiers (testing data).

	TH	TP	FP	FN	TN	SN (%)	SP	PP	ACC	AUC
RFC	0.077	223	960	147	3804	60.27	79.85	18.85	78.44	77.59
ADB	0.487	234	928	136	3836	63.24	80.52	20.14	79.28	**79.25**
GNB	0.065	258	1396	112	3368	69.73	*70.7*	*15.6*	70.63	78.08
KNNC	0.065	219	892	151	3872	*59.19*	81.28	19.71	79.68	72.11
SVC	0.065	221	1380	149	3384	59.73	71.03	13.8	*70.22*	*71.51*
DNN	0.13	238	688	132	4076	64.32	85.56	25.7	84.03	**83.48**

TH, threshold; RFC, random forest classifier; ADB, AdaBoost classifier: GNB, Gaussian naive Bayes; KNNC, K-nearest neighbor classifier; SVC, support vector machine; DNN, deep neural network.

**Table 4 ijerph-16-01876-t004:** Comparison of the performance for six classifiers (10-fold cross-validation).

Classifier	AUC	Classifier	AUC
RFC	79.4	ADB	**80.5**
GNB	80.0	KNNC	72.2
SVC	69.7	DNN	**83.5**

**Table 5 ijerph-16-01876-t005:** Correlation coefficients of the variables (over 0.09) among 11 variables.

Variable	Corr. Coff.	Variable	Corr. Coff.
Brain surgery required	*0.124062*	Admission mode	*−0.093137*
Stroke type	**0.203716**	Mortality	1

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
