# Peer review of "The Use of Deep Learning to Predict Stroke Patient Mortality"

_ijerph, 2019, doi:10.3390/ijerph16111876_

Reviewer 1 Report

The article proposes to use a deep neural network to detect stroke using medical service use and health behavior data. The results of experiments on the dataset of 15,009 patients with stroke are presented.

Comments:

1. Clearly formulate the novelty of your approach in the introduction section of the paper. What is the context of application (e-health, assisted living systems, etc.)?

2. Introduction section: discuss the advantages and limitations of various methods (including the ones that are not based on deep learning such as heuristic/nature-inspired) for stroke prediction,. The following works on stroke and related heart/cardiovascular disease forecasting highly relevant to your research should be discussed and cited:

Teoh D. (2018). Towards stroke prediction using      electronic health records. BMC medical informatics and decision making,      18(1), 127. doi:10.1186/s12911-018-0702-y

Beritelli, et al. (2018). A novel training method      to preserve generalization of RBPNN classifiers applied to ECG signals      diagnosis. Neural Networks, 108, 331-338. doi:10.1016/j.neunet.2018.08.023

Pereira, et al. (2018). Stroke lesion detection      using convolutional neural networks. Proceedings of the International      Joint Conference on Neural Networks, doi:10.1109/IJCNN.2018.8489199

Wu et al. (2019). Risk assessment of hypertension      in steel workers based on LVQ and fisher-SVM deep excavation. IEEE Access,      7, 23109-23119. doi:10.1109/ACCESS.2019.2899625

3. Provide more statistical information about patients: mean age, gender distribution, etc.

4. Section 3.3: did you use cross-validation? If not, why?

5. Figure 2: why you used all 11 principal components (PCs) from PCA? Ussually, only a few PCs, which explain most variability in the data are used.

6. Explain class 0, class 1 and class 2.

7. Section 3.: add a figure presenting an architecture of DNN.

8. Table 3: correlation coefficients are low and do not seem to be explanative. Suggest removing the table as irrelevant.

9. Conclusions section also should summarize main numerical results of the experiments.

Author Response

Responses to the Reviewers’ Comments (ijerph-506161)

We thank the editor and reviewers for their careful and thorough review. The comments and suggestions were very useful and helped to sharpen our thinking during the revision process. We have made the following major changes to the manuscript based on the reviewer's suggestions.

In our original manuscript, we added related work section for comparing with previous literatures to check the novelty of our proposed method. We carefully read and reviewed thirteen papers in regards to the state-of-art research [23-26; 32-41] on machine learning and deep learning methods between 2013 to 2019.

Following the suggestions by Reviewer 1, we added the 10-fold cross-validation results (Table 4) and Figure 4 about the architecture of the proposed DNN in the Section 3.5.

We added Table 1 and modified Table 5 to explain the variables better.

Other changes will be elaborated in our responses to reviewers’ comments.

Reviewer 2 Report

The research article entitled “The Use of Deep Learning to Predict Stroke Patient Mortality” is a comprehensive description of deep neural network (DNN) machine learning approach for efficient detection of stroke from the available medical service use and health behavior data.

Following minor comments should be considered by the authors to improve the content description:

1.       In the introduction, line number 42, “Deep learning using big data has been employed to predict disease, [17-19].” comma should be removed after disease.

2.       In Fig. 2, the text inside the box need to be corrected. The text above the arrows is overlapping inside the box space and should be aligned with enough spacing.

3.       In Fig. 3 “Two-dimensional plots of the first and second principal components”, the boxes of (c) and (d) should be aligned at the same level. In material and methods section 3.4: Preprocessing, appropriate description about the class 0, class 1 and class 2 as shown in Fig. 3 is lacking and should be included in the figure description.  

4.       In Table 2, (Confusion matrix values) the text in the columns need to be aligned within the column borders.

5.       In the Result and Discussion, line number 183, “Table 2 lists the correlation coefficients 181 (r-values) of the 11 variables; the best correlations (+0.2) are highlighted in bold and two other 182 informative results (+-0.1) are in italics and shown in red.”. Here, (+-0.1) should be written as

 (+/- 0.1).

6.       In the conclusion paragraph, line number 208 “In future, we will modify and apply our method to analyses of other medical service use and 208 health behavior datasets on conditions such as dementia.” should be replaced with “modify and apply our method for the analysis of other medical service use

7.       Reference number 42 appears incomplete and all relevant details should be included.

Author Response

(The authors gave the same response as above.)

Reviewer 3 Report

This paper shows the use of component analysis and machine learning methods to automatically predict the mortality of stroke patients. Using a seemingly limited set of variables, the authors attained a PPV of 25.7% and AUC of 83.5% with a deep neural network.

Although the paper is a good read and shows relevant results, a fair number of oddities were found while reading it.

The original features, as well as their specifications, are unclear in various places. Section 3.3 (Methods) describes the use of 11 variables, yet only 9 attributes were enumerated in that section. It would be important to describe how these attributes amount to 11 variables, and/or add any missing attributes if that is the case. Section 3.2 shows what could be the two variables missing, the hospital region and the hospital’s number of beds. Are these features reliable for stroke detection? Moreover, the descriptions for the type of insurance, admission mode, and type of stroke could be improved, as it may not be clear exactly which categories are assumed in each variable.

Section 3.4 describes that an overcomplete PCA step was applied over the 11 features, resulting in 24 continuous variables. However, Figure 2 shows 11 as the number of PCA components. Is this a typo? In addition, the choice of 24 as the number of components is not justified in the text. It is particularly confusing because PCA is formerly explained as a way to “reduce dimensions” rather than increase them.

Although Section 3.3 presents a train-test split, Section 4 later explains that the training set was again split for validation, subsequently being used for identifying an optimal neural network architecture. This seems to be part of the methods rather than the results. It would be best to move this to Section 2 and the assume this 3-part separation in the rest of the pipeline (i.e. including it in Figure 2), so as to better understand the role of this validation set.

The description of the neural network was a bit thin in places. A regularization technique is mentioned (lines 140 and 143), but not fully described. What was the learning rate of the optimizer described in lines 154-156? The preference of accuracy as the loss function over the usual binary cross entropy should also be explained further. At some point, the paper indicates the optimal threshold for each classifier, but the method employed to identify this threshold were not disclosed.

Minor issues:

line 42: “has been employed to predict disease , [17-19]” a comma before a list of citations is unusual.

line 116: missing period in “respectively(.) The Sn,”

line 171: the rows of Table 2 are not well aligned in the provided document

line 172: do the authors mean ABC rather than ADB?

Author Response

(The authors gave the same response as above.)

Reviewer 4 Report

Authors concern on a valuable topic of CAD method for stroke modality prediction. The paper is organized well and the results can demonstrate the efficiency of the proposed method. However, as we all know, deep learning makes great breakthroughs in CAD system for diseases. The authors miss some related papers and should add some related references for cancers classification and tumor detection, such as:

Jiao, Zhicheng, et al. "A parasitic metric learning net for breast mass classification based on mammography." Pattern Recognition 75 (2018): 292-301.

Hu, Yang et al. "Mammographic Mass Detection Based on Saliency with Deep Features." Proceedings of the International Conference on Internet Multimedia Computing and Service. ACM, 2016.

Yang, Dongxiao et al. "Asymmetry Analysis with Sparse Autoencoder in Mammography." Proceedings of the International Conference on Internet Multimedia Computing and Service. ACM, 2016.

Shen, Dinggang, Guorong Wu, and Heung-Il Suk. "Deep learning in medical image analysis." Annual review of biomedical engineering 19 (2017): 221-248.

Suk, Heung-Il, and Dinggang Shen. "Deep learning-based feature representation for AD/MCI classification." International Conference on Medical Image Computing and Computer-Assisted Intervention. Springer, Berlin, Heidelberg, 2013.

Hua, Kai-Lung, et al. "Computer-aided classification of lung nodules on computed tomography images via deep learning technique." OncoTargets and therapy 8 (2015).

Anthimopoulos, Marios, et al. "Lung pattern classification for interstitial lung diseases using a deep convolutional neural network." IEEE transactions on medical imaging 35.5 (2016): 1207-1216.

Kumar, Devinder, Alexander Wong, and David A. Clausi. "Lung nodule classification using deep features in CT images." 2015 12th Conference on Computer and Robot Vision. IEEE, 2015.

Jiao, Zhicheng, et al. "A deep feature based framework for breast masses classification." Neurocomputing 197 (2016): 221-231.

Jiao, Zhicheng, et al. "Deep Convolutional Neural Networks for mental load classification based on EEG data." Pattern Recognition 76 (2018): 582-595.

Author Response

Responses to the Reviewers’ Comments (ijerph-506161)

Point-by-point response to Reviewer 4's comments

Overall comment: Authors concern on a valuable topic of CAD method for stroke modality prediction. The paper is organized well and the results can demonstrate the efficiency of the proposed method. However, as we all know, deep learning makes great breakthroughs in CAD system for diseases. The authors miss some related papers and should add some related references for cancers classification and tumor detection, such as:

Jiao, Zhicheng, et al. "A parasitic metric learning net for breast mass classification based on mammography." Pattern Recognition 75 (2018): 292-301.

Hu, Yang et al. "Mammographic Mass Detection Based on Saliency with Deep Features." Proceedings of the International Conference on Internet Multimedia Computing and Service. ACM, 2016.

Yang, Dongxiao et al. "Asymmetry Analysis with Sparse Autoencoder in Mammography." Proceedings of the International Conference on Internet Multimedia Computing and Service. ACM, 2016.

Shen, Dinggang, Guorong Wu, and Heung-Il Suk. "Deep learning in medical image analysis." Annual review of biomedical engineering 19 (2017): 221-248.

Suk, Heung-Il, and Dinggang Shen. "Deep learning-based feature representation for AD/MCI classification." International Conference on Medical Image Computing and Computer-Assisted Intervention. Springer, Berlin, Heidelberg, 2013.

Hua, Kai-Lung, et al. "Computer-aided classification of lung nodules on computed tomography images via deep learning technique." OncoTargets and therapy 8 (2015).

Anthimopoulos, Marios, et al. "Lung pattern classification for interstitial lung diseases using a deep convolutional neural network." IEEE transactions on medical imaging 35.5 (2016): 1207-1216.

Kumar, Devinder, Alexander Wong, and David A. Clausi. "Lung nodule classification using deep features in CT images." 2015 12th Conference on Computer and Robot Vision. IEEE, 2015.

Jiao, Zhicheng, et al. "A deep feature based framework for breast masses classification." Neurocomputing 197 (2016): 221-231.

Jiao, Zhicheng, et al. "Deep Convolutional Neural Networks for mental load classification based on EEG data." Pattern Recognition 76 (2018): 582-595.

Response: We sincerely thank you for your time and efforts in reading and commenting on our paper. We have added the related references in the Section II as follows:

“Especially, there have been many computer-aided diagnosis systems using deep learning for diseases [28-37].”

Round  2

Reviewer 1 Report

The authors have addressed all my concerns and comments, and did a good job in revising the article. I have no further comments. I recommend the article for acceptance.